# A Strategy for Hospital Pharmacists to Control Antimicrobial Resistance (AMR) in Japan

**DOI:** 10.3390/antibiotics10111284

**Published:** 2021-10-21

**Authors:** Yukihiro Hamada, Fumiya Ebihara, Ken Kikuchi

**Affiliations:** 1Department of Pharmacy, Tokyo Women’s Medical University Hospital, Tokyo 162-8666, Japan; ebihara.fumiya@twmu.ac.jp; 2Department of Infectious Disease, Tokyo Women’s Medical University Hospital, Tokyo 162-8666, Japan; kikuchi.ken@twmu.ac.jp

**Keywords:** antimicrobial stewardship, antifungal stewardship, strategy of pharmacist, therapeutic drug monitoring, pharmacokinetics/pharmacodynamics

## Abstract

In Japan, there is concern regarding the relation between the inappropriate use of antibiotics and antibiotic resistance (AMR). Increased bacterial resistance is due in part to the inappropriate use of antimicrobial agents. The support of the pharmacist becomes important, and there is growing interest in antimicrobial stewardship to promote the appropriate and safe use of antimicrobials needed for the optimal selection of drugs, doses, durations of therapy, therapeutic drug monitoring (TDM), and implementations of cost containment strategies in Japan. Pharmacists should strive to disseminate the concept of “choosing wisely” in relation to all medicines, implement further interventions, and put them into practice. In this article, we present data for antimicrobial stewardship and Japan’s AMR action plan, focusing on how pharmacists should be involved in enabling physicians to choose antimicrobials wisely.

## 1. Introduction

The spread of drug-resistant bacteria poses a serious threat to human beings worldwide [1]. At the 2015 World Health Assembly, the World Health Organization appointed each country to develop a national action plan (NAP) to counter the rise in antimicrobial resistance (AMR) [2]. As of 2018, 115 countries and territories had implemented such plans, with 34 others in the process of developing their own proposals [3]. Japan announced its National Action Plan on Antimicrobial Resistance in April 2016, setting specific goals for a reduction in antimicrobial use by 2020 [4]. There is a background of Japan’s inordinately frequent use of broad-spectrum oral antibiotics [5]; its NAP set reduction targets for selected antibiotic categories in addition to total antimicrobial use. These goals included a 33% reduction in total antimicrobial use; 50% reductions in oral fluoroquinolone, cephalosporin, and macrolide use; and a 20% reduction in parenteral antimicrobial use in 2020 relative to 2013. In our hospital research conducted over a period of 2 years, we found significant reductions (*p* < 0.01) in the defined daily doses (DDDs) of oral third-generation cephalosporins and fluoroquinolones. Furthermore, a significant reduction was also found in the total DDDs of the antibiotics used. In addition, the implementation of antimicrobial stewardship reduced the unnecessary antimicrobial prescription rate for acute respiratory tract infections and acute diarrhea (Figure 1) [6].

Infectious disease (ID) pharmacists have to extend their support for the appropriate use of antimicrobials prescribed by attending physicians to not only hospitalized patients but also outpatients. “Choosing wisely”, the United States-based health education campaign, warns against the unnecessary use of antimicrobials [7]. Pharmacists are encouraged to promote the concept of “choosing wisely” in relation to other medicines, implement further interventions, and put them into practice. AMR Alliance Japan also suggests that efforts should be made to promote an increase in the number of healthcare workers trained to conduct therapeutic drug monitoring (TDM) in the Japanese government’s next NAP on AMR, which will be implemented as of 2021 on the basis of a policy recommendation published in July 2019 [8]. In this article, we present data for antimicrobial stewardship in Japan, focusing on methicillin-resistant *Staphylococcus aureus* (MRSA), candidiasis, and extended-spectrum β-lactamase (ESBL)-producing *Enterobacteriaceae*, which are particular problems in hospitals, and we provide suggestions on how pharmacists should practice to enable physicians to choose antimicrobials wisely.

## 2. Antimicrobial Stewardship with MRSA and Extended-Spectrum β-Lactamase (ESBL)-Producing *Enterobacteriaceae*

### 2.1. Epidemiology of MRSA in Japan

In Japan, the most common AMR organism is MRSA [9]. MRSA is one of the most common bacteria causing healthcare-associated infections and is the most notable resistant bacterium in hospitals.

According to the Japan Nosocomial Infection Surveillance (JANIS) sponsored by the Ministry of Health, Labour, and Welfare, MRSA was isolated from 2162 out of 2167 medical institutions (99.8%) surveyed in 2020. MRSA previously accounted for 50 to 70% of *Staphylococcus aureus* isolates from hospitalized patients, but, in recent years, there has been a decreasing trend [10]. Tsuzuki et al. [11] reported that the number of bloodstream infection (BSI) deaths attributable to *S. aureus* was expected to be 17,412 in 2011 and 17,157 in 2017, respectively, out of the whole population (126.8 million) in Japan. Among them, the cases attributed to MRSA accounted for 5924 (34.0%) in 2011, and this figure decreased to 4224 (24.6%) cases in 2017. The number of BSI deaths attributable to MRSA has shown a decreasing trend. Tsuzuki et al. [12] estimated the disease burden of BSI caused by major Japanese antimicrobial-resistant bacteria between 2015 and 2018 in terms of disability-adjusted life-years (DALYs; [13]). The disease burden due to MRSA in Japan in 2018 was estimated to be 57.8 DALYs per 100,000 population, which was reported to be significantly higher in Japan than in EU/EEA countries (57.8 vs. 20.9 per 100,000 population) [14].

Uematsu et al. [15] analyzed the clinical and economic burdens attributable to methicillin resistance in *S. aureus* in Japanese hospitals. The participants were inpatients from admission on 1 April 2014 to discharge on 31 March 2016. A total of 7188 patients were classified into the MRSA group and 7717 patients into the MSSA group. The outcomes were assessed by the length of hospital stay, inpatient costs, and in-hospital mortality, and the disease burden was compared. The MRSA-group-adjusted effects were 1.03-fold (95% confidence interval (CI), 1.01–1.05) and 1.04-fold (95% CI, 1.01–1.06), with an odds ratio of 1.14 (95% CI, 1.02–1.27). This result shows that the disease burden was higher in the inpatients with MRSA than in those with MSSA; moreover, the attributable burden of methicillin resistance was significant after adjustments.

In Japan, vancomycin, teicoplanin, arbekacin, and linezolid had been used as four therapeutic agents against MRSA. Recently, newer anti-MRSA agents have been approved, including daptomycin in 2011 and tedizolid in 2018 [16]. As a result, it was found that the total anti-MRSA agent use was increased significantly over time. It is thought that this may reflect compliance with the guidelines, the launch of new anti-MRSA drugs, and less promotion of the original agents in Japan. Goto et al. [17] reported on the trends of only anti-MRSA agent use in Japan based on sales data. The total anti-MRSA agent use was found to have significantly increased from 2006 to 2015 (*p* for trend = 0.027). The individual trends for arbekacin and teicoplanin showed decreases, while vancomycin, daptomycin, and linezolid use showed significant increases. In addition, there was no significant change in the use of intravenous linezolid, but the use of oral linezolid increased significantly. As a result, the ratio of oral linezolid use to total linezolid use increased from 25.5% in 2006 to 39.9% in 2015. Therefore, we are paying close attention to the amount of oral linezolid used in Japan. On the other hand, arbekacin and teicoplanin use decreased while vancomycin use increased following the launch of generic medicines. These results may reflect compliance with the guidelines, the launch of new anti-MRSA agents, and less promotion of the original agents.

In particular, vancomycin, teicoplanin, and arbekacin were difficult to achieve the target trough concentration with the dose setting in the Japanese package insert. The adequate dosage in compliance with the guidelines may be one factor in the increase in the dosage. The AMR problem might be even more serious in Japan if the dosage was the same as before.

### 2.2. Antimicrobial Stewardship of MRSA by Pharmacists in Japan

As mentioned earlier, the frequency of MRSA isolation has decreased in Japan in recent years, but MRSA remains the most frequently isolated resistant pathogen. Therefore, the first priority in MRSA infection control is to prevent the spread of MRSA [18], and the thorough implementation of standard precautions, such as hand hygiene, is an effective means of preventing the spread of MRSA in hospitals [19,20,21]. In addition to measures to prevent the spread of MRSA, the appropriate use of antimicrobial agents is necessary to reduce the risk of MRSA infection. It is desirable to take a proactive approach as an organization by adopting antimicrobial stewardship and collaborating with multiple professions [22,23].

Niwa et al. [24], Japanese pharmacists, reported that antimicrobial stewardship interventions are effective in reducing the inappropriate use of antibiotics, lowering MRSA ratios, shortening hospital stays, and reducing healthcare costs. The prolonged use of antibiotics over 2 weeks was significantly reduced after the active implementation of antimicrobial stewardship (2.9% vs. 5.2%, *p* < 0.001). A significant reduction in the antimicrobial consumption was observed in aminoglycosides (*p* < 0.001), carbapenems (*p* = 0.003), and second-generation cephalosporins (*p* = 0.03), leading to an 11.7% reduction in the antibiotic costs. The rate of MRSA decreased significantly from 47.6% to 39.5% (*p* = 0.026). Moreover, the average length of hospital stay was shortened by 2.9 days after the active implementation of antimicrobial stewardship.

Ohashi et al. [25], Japanese pharmacists, reported on the impact of antimicrobial stewardship by pharmacist intervention in MRSA bacteremia patients using a treatment bundle. With the intervention of an antimicrobial stewardship team pharmacist, an increase was observed in the appropriate duration of therapy, incidences of the early use of anti-MRSA drugs, and the number of negative follow-up blood cultures, and a decrease was observed for 30-day mortality and hospital mortality, with significant differences. A multivariate analysis of these indicated that the intervention group was independent of 30-day mortality and hospital mortality risk reduction factors (odds ratio (OR), 0.33; 95% confidence interval (CI), 0.12–0.86, and OR, 0.20; 95% CI, 0.07–0.53).

In addition to the selection of appropriate antimicrobial agents and the duration of the treatment, the implementation of TDM for anti-MRSA drugs is essential for antimicrobial stewardship.

The “Therapeutic monitoring of vancomycin for MRSA infections: A revised consensus guideline” [26] published by several societies, including the Infectious Diseases Society of America, also sets a target for dosing guided by the target area under the curve (AUC) over 24 h for clinical efficacy and safety and states that evaluation via trough concentration-guided dosing alone is not recommended in severe MRSA infections. The Japanese TDM guidelines for antimicrobial agents will be revised in 2021, and evaluation by AUC-guided dosing will be recommended.

Our group, Suzuki et al. [27], reported that the risk factors for vancomycin-induced nephrotoxicity were comparable in both trough concentration and/or the AUC. The incidence of nephrotoxicity can be reduced by controlling vancomycin via trough concentration- and AUC-guided dosing as well as promoting antimicrobial stewardship. Clearly, the AUC-guided dosing of vancomycin is important. However, accurate AUC-guided dosing may require multiple blood samples. Therefore, it is important to properly use AUC and/or trough monitoring. AUC-guided dosing is recommended for patients at a high risk of renal impairment, but, for patients without AUC-guided dosing, it is important to build a predictable model even with conventional trough monitoring. Especially under COVID-19, we want to reduce blood sampling points as much as possible, and this result will be one tool for pharmacists to propose TDM to physicians.

In a multicenter study, Hashimoto et al. [28] identified risk factors for early- and late-phase vancomycin-induced acute kidney injury (AKI) to identify candidates for AUC-guided dosing, rather than trough concentration (Cmin)-guided dosing, who require a more accurate dose titration to reduce the risk of AKI. AKI was observed in 8.5% of the patients (159/1882). AKI appeared within the first 7 days of therapy (early phase) in the majority of the patients. Important AKI risk factors during the early phase were identified as a Cmin > 20 mg/L, concurrent diuretic or piperacillin/tazobactam use, intensive care unit stay, and pre-existing renal dysfunction. A transiently elevated Cmin (>15–20 mg/L) was not associated with a greater risk of AKI. In patients with the risk factors for AKI, the cut-off Cmin and the estimated safe Cmin for reduced AKI risk were 18.8–21.0 mg/L and <11.7–13.5 mg/L, respectively. As a result, the patients with AKI risk factors require a low target Cmin. The presence of several risk factors may indicate a need for more accurate dose titration using AUC-guided administration.

Our pharmacists should strive to reduce the patient risk while creating novel evidence for TDM.

### 2.3. Anti-Extended-Spectrum β-Lactamase (ESBL)-Producing Enterobacteriaceae Agents Stewardship

#### Epidemiology of ESBL in Japan

The use of broad-spectrum antibiotics has increased infections caused by antimicrobial-resistant bacteria. The epidemic of these resistant bacteria is a global threat to public health systems [29]. Among these, ESBL-producing *Escherichia coli* (ESBL *E. coli*) infections represent the greatest threat [30,31], with the number of infected patients increasing around the world [32,33], especially in Asia, Africa, and Latin America [34]. In addition to its use for humans in clinical practice, far greater amounts of antimicrobials are currently consumed by the animal husbandry sector, including aquaculture. This massive usage is not only for the treatment or prophylaxis of food animals but also for growth promotion purposes [35].

Nishiyama et al. [36] reported on wastewater samples from a municipal wastewater treatment plant and hospital wastewater for six species of antibiotic-resistant bacteria: ESBL-producing *Enterobacteria,* carbapenem-resistant *Enterobacteria* (CARBA), multidrug-resistant *Acinetobacter* (MDRA), multidrug-resistant *Pseudomonas aeruginosa* (MDRP), methicillin-resistant *Staphylococcus aureus* (MRSA), and vancomycin-resistant *Enterococci* (VRE). They registered a high percentage of antibiotic-resistant bacteria in the municipal wastewater treatment plant samples (>66%) for all the antibiotic-resistant bacteria except for MDRP, indicating a high prevalence in the population. The proportion of hospital wastewater samples was low (<78%), and VRE was not detected throughout the study. CARBA and ESBL were detected in all the wastewater samples, and MDRA and MRSA were very abundant in Japan.

ESBLs mediate resistance to third-generation cephalosporins in *E. coli* and *Klebsiella pneumoniae*. In general, serious infections caused by these strains are treated with carbapenems. However, if carbapenem is used in all cases, carbapenem resistance may be selected.

The MERINO open-label randomized controlled study has provided clear evidence that piperacillin–tazobactam should be avoided for the targeted therapy of bloodstream infections due to ESBL-producing *E. coli* and *K. pneumoniae*, regardless of the patient population, source of infection, bacterial species, and susceptibility result of piperacillin–tazobactam. The verification of other treatments for mild infections with ESBL may not yet be sufficiently validated [37]. Data on the clinical efficacy of new drugs specific to infections caused by carbapenem-resistant pathogens are gradually being published. It seems to favor newer drugs rather than previously available treatments.

As more treatment options become widely available for carbapenem-resistant, Gram-negative infections, the role of antimicrobial stewardship will become crucial in ensuring the appropriate and rational use of these new agents [38].

Postmenopausal women in Japan have an increased proportion of ESBL *E. coli* and lower susceptibility to LVFX. The ESBL *E. coli* isolates appeared to have a high susceptibility to tazobactam–piperacillin, cefmetazole, carbapenems, aminoglycosides, and fosfomycin [39].

Nakai et al. [40] reported that the problem of ESBL production includes not only nosocomial infections but also community-acquired infections. Antimicrobial usage (for more than 4 days) during the preceding 60 days was a risk factor, especially the usage of aminoglycoside, oxazolidinone, tetracycline, fluoroquinolone, trimethoprim/sulfamethoxazole, and second- and fourth-generation cephalosporin [40]. Japanese pharmacists need to consider the background and treatment strategies.

### 2.4. Antimicrobial Stewardship of ESBL by Pharmacists in Japan

Pharmacists play an important role in antimicrobial stewardship, including optimizing the treatment of multidrug-resistant pathogens. It is estimated that up to 50% of hospital prescriptions for antibiotics are unnecessary [41]. Recently, the development of antibiotics has slowed considerably, and the options for treating increasingly resistant infections have become increasingly limited [42]. This is also the case in Japan. It is important for the pharmacist to recognize ESBL-producing pathogens and understand which antibiotics are supported by the strongest data and outcomes. The pharmacist should be aware of the source of infection and the treatment options available. Additionally, pharmacists can implement antimicrobial stewardship throughout the healthcare system, such as avoiding the unnecessary administration of antibiotics and de-escalating use as soon as possible, which can help prevent patients from developing resistant pathogens. Antimicrobial stewardship can provide pharmacists with the information they need to avoid the overuse of antibiotics and to help prevent the development of forms of antimicrobial resistance, such as ESBLs [43].

Kusama et al. [44] reported that the National Action Plan on Antimicrobial Resistance significantly reduced antimicrobial use in Japan. This plan resulted in both an immediate and accelerated reduction in antimicrobial use. Antimicrobial use reductions of 15.0% for total antimicrobials, 26.3% for cephalosporins, 24.6% for macrolides, and 23.5% for fluoroquinolones were predicted for 2020 relative to 2013. While Japan’s National Action Plan has contributed to the reduction in national antimicrobial use over the past 5 years, sustainable action is still needed to continue to improve antimicrobial stewardship and promote countermeasures to antimicrobial resistance.

As reported by a group of Japanese pharmacists, Kato et al. [45], a combination of prescription support activities and oral antimicrobial treatment reporting systems is an effective way to facilitate the use of appropriate antimicrobial agents. These authors evaluated preauthorization, a prospective audit, and a feedback system. The total annual amount of oral third-generation cephalosporin usage decreased significantly over time between the phases. During the same period, the incidence rate of MRSA, ESBLs, and AmpC β-lactamase (AmpC)-producing bacteria was not changed significantly, indicating that oral third-generation cephalosporin usage was reduced without a concomitant increase in the drug-resistant bacteria.

The rate of urinary tract infections caused by ESBLs has increased worldwide. The development of new antibiotics that are effective against ESBL *E. coli* is undoubtedly a top health care priority by the World Health Organization [46]. The cefmetazole of cephamycin is stable against the hydrolysis by ESBLs and has robust in vitro activity with lower minimum inhibitory concentrations (MICs) against ESBL *E. coli* isolates [47]. Therefore, cefmetazole is receiving increasing interest as a potential carbapenem-saving treatment option for ESBL infections. Cefmetazole and flomoxef are commonly used in Japan as they are expected to have therapeutic effects [48,49]. As the evidence, a few observational studies have examined the efficacy of cephamycin against ESBL *E. coli* infection. Doi et al. reported that the efficacy of cefmetazole against ESBL *E. coli* for UTI was similar to that of carbapenems (90%) without major adverse events [50]. Another study included patients with ESBL *E. coli* bacteremia who were treated with a carbapenem (*n* = 43) or cefmetazole (*n* = 26) and reported comparable mortality rates in the two groups [51]. In both studies, the majority (>90%) of ESBL *E. coli* included in the analyses comprised ESBLs. Unfortunately, the dosage and usage of cefmetazole was not described in the reports.

Therefore, Hamada et al. [52], our group of Japanese pharmacists, retrospectively evaluated the dose adjustment by renal function with the administration of various doses. The extension in infusion time from 0.5 h to 1 h is a feasible practice in clinical settings and recommended strategy when administering cefmetazole. In conclusion, 1 g of cefmetazole infused for over 1 h every 8 h proved to be efficacious for the treatment of UTI caused by ESBL *E. coli* with cefmetazole MIC ≤ 4 mg/L in PK/PD theory. This PK/PD relationship finding can be applied as a foundation for future clinical studies addressing the utility of cefmetazole or other antimicrobial agents as a carbapenem-sparing treatment option for ESBL *E. coli* infections.

The strategies of these pharmacists can play an essential role in assisting providers with the selection of appropriate antimicrobial agents for these multidrug-resistant pathogens in order to improve patient outcomes.

## 3. Antifungal Stewardship

### 3.1. Epidemiology of Fungal Infections in Japan

In recent years, the prevalence of invasive candidiasis has increased with the advancement of medicine, representing an important complication. In the United States, Candida is now the fourth most common pathogenic microorganism in nosocomial bloodstream infections [53]. Although there are few epidemiological studies in Japan, the breakdown of blood borne isolates reported by the JANIS shows that the most common *Candida albicans* of the *Candida species* was reported to be tenth [9]. The Epidemiological Investigation Committee for Human Mycoses in Japan included a total of 328,318 blood cultures from 2003 to 2014 according to a retrospective epidemiological survey of candidemia and causative Candida species. The prevalence of fungi in all the cultures and in the positive cultures were 0.58% and 4.46%, respectively, and the frequency of *C. albicans* has significantly decreased, while that of *C. glabrata* has increased during the last 6 years in Japan [54].

Kishimoto et al. [55] reported the overall epidemiology of invasive fungal disease (IFD) in Japanese children. The proven or probable of IFD was 26 of 20,079 hospitalized patients (0.13%) from 2011 to 2015. The overall mortality was 23%, and the attributable mortality of IFD was 12%. These results warn of the emergence of non-albicans Candida species as important pathogens in pediatric IFD.

In 20 years, the number of treatment options for invasive fungal infections has increased in Japan. Antifungal agents, such as echinocandins, voriconazole, liposomal amphotericin B, and posaconazole, which are indicated for invasive fungal infections, have been launched in Japan. As a result, the use of antifungal drugs for invasive fungal infections has been on the rise. It is thought that this is related to the increasing number of hosts with low immunity in Japan. Kawabe et al. [56] provided a detailed description of the trend of antifungal drugs used in Japan from 2006 to 2015 based on sales data as an alternative indicator of the trends in fungal infections. The total amount of antifungal drugs used decreased over time, and the main reason for this result was the decreasing use of antifungal drugs for superficial fungal infections. Interestingly, the use of antifungal agents for invasive fungal infections increased, and the total use of antifungal agents decreased (r = −0.057, *p* for trend < 0.0001). Oral and parenteral antifungals were significantly decreased by 44.1% (r = −0.056, *p* for trend < 0.0001) and 27.1% (r = −0.0012, *p* for trend = 0.00061), respectively. The use of antifungal agents for superficial fungal infections was significantly decreased to 49.8% (r = −0.061, *p* for trend < 0.0001). However, the use of antifungal agents for invasive fungal infections increased significantly to 19.9% (r = 0.0032, *p* for trend = 0.00045). Among the parenteral antifungal drugs used for invasive fungal infections, echinocandins accounted for more than half.

### 3.2. Antifungal Stewardship by Pharmacists in Japan

The main objective of the antifungal stewardship programs (AFSPs) is to optimize the use of antifungal drugs by integrating the experience and knowledge of professionals to address issues that impede the appropriate use of antifungal drugs [57].

Hamdy et al. [58] offered suggestions for both process metrics (i.e., those that measure the effect of an intervention on antifungal use) and outcome metrics (i.e., those that measure the effect of an intervention on the resistance patterns and clinical outcomes) for antifungal stewardship. Table 1 was partially modified by the authors on the basis of these suggestions. Pharmacists play an important role in AFSPs, including optimizing the treatment in antifungal therapy.

Kawaguchi et al. [59] reported that the patients who received systemic antifungal therapy from 2011 to 2016 were divided into pre-intervention and intervention groups, and the monthly average number of treatment days was significantly lower in the intervention group (15.1 ± 3.1 vs. 12.7 ± 4.3, *p* = 0.009), and the cost of the antifungal therapeutics decreased by USD 260,520 (13.5%) over three years. In addition, there was a downward trend in both the 30-day mortality and in-hospital mortality among the patients with candidiasis. The increased selection of antifungal agents by the AFSP interventions conducted in Japanese hospitals has resulted in lower antifungal drug usage and cost savings, demonstrating a tendency to improve the prognosis of the patients with candidemia.

As reported by a group of Japanese pharmacists, Samura et al. [60], the process parameter cumulative optimal use of antifungal drugs was significantly increased in the post-AFSP group (*p* = 0.025). In addition, the median number of days of antifungal treatment was 6.0 (interquartile range (IQR) 0.3–15.7) and the median 3.4 (IQR 1.9–3.4) per 1000 patient days, with a significant decrease in the post-AFSP group (*p* < 0.001). The outcome parameter expenditure on antifungal drugs was USD 9390.5 ± 5687.1 and USD 5930.8 ± 4687.0 before and after AFSPs, respectively, with a significant decrease in the post-AFSP group (*p* = 0.002).

We investigated the compliance with the standard dosage of voriconazole in our hospital (Figure 2) [61]. The data from an overall total of 118 patients who received voriconazole in our hospital from April 2015 to March 2019 were included. Of these, 26 patients whose voriconazole use was prophylactic were included, and 92 patients whose voriconazole use was therapeutic were included. The appropriate dose of voriconazole was a loading dose of 5 to 6 mg/kg twice daily, followed by a maintenance dose of 3 to 4 mg/kg twice daily. In the patients with hepatic dysfunction (Child–Pugh A–C), the maintenance dose was reduced to 1.5 to 2 mg/kg. Since the dose was calculated based on body weight, it was considered appropriate to round the dose to within 10% of the recommended dose. Although the difference was not significant due to the small sample size, the rate of adherence to the standard dose of voriconazole in the prophylactic dose and the loading dose and maintenance dose in the treatment all improved as a result of the intervention of a pharmacist.

AMR Alliance Japan, a policy proposal announced in July 2019, proposes that items will be added, changed, or emphasized in the Japanese government’s next National Action Plan on Antimicrobial Resistance to be implemented in 2021 [7]. The efforts are encouraged in this regard to promote an increase in the number of trained healthcare professionals to perform TDM. The TDM of voriconazole of the antifungal agent is associated with efficacy and safety, and pharmacists play a particularly important role in TDM [62,63]. Voriconazole concentration is associated with adverse effects, including visual symptoms, neuropathy, and hepatotoxicity [64,65,66,67,68,69].

Hamada et al. [70] used a previously unreported meta-analysis for the first time to investigate the optimal concentration of voriconazole. This result was published as a practical guideline for voriconazole TDM [71] and has been extensively clinically applied following a consensus review conducted by the Japan Society of Chemotherapy and the Japanese Society for Therapeutic Drug Monitoring. Subsequently, Hamada et al. [72] reported that, by observing the concentration with TDM of voriconazole, the pharmacist suggested a change in treatment after the onset of AEs, and most patients completed the treatment. This study enrolled a relatively large number of patients, 401, at five hospitals in Japan. Hepatotoxicity and visual symptoms were significantly correlated with the voriconazole trough concentration at the onset of AEs, and the trough cut-offs were 3.5 μg/mL for hepatotoxicity and 4.2 μg/mL for visual symptoms. As a great result, with the voriconazole dose adjustment based on TDM, the treatment was completed in 88.9% of the patients with hepatotoxicity and 96.4% of the patients with visual symptoms.

In our latest study, Hanai et al. [73] conducted a systematic review and meta-analysis to assess the relationship between the voriconazole trough concentration and the efficacy and safety, and re-evaluated it in 2554 adult patients. Although further high-quality trials are needed, our findings recommend an appropriate target voriconazole trough concentration of 1.0–4.0 μg/mL for increased clinical success while minimizing toxicity. On the other hand, pediatric data are also important. However, data on the target range of voriconazole concentrations in children are lacking because most studies are limited by small sample sizes and other factors. Hanai et al. [74] used a similar approach to suggest that the target concentration range for voriconazole in children, particularly for Asian populations, is 1.0–3.0 μg/mL.

As a recent topic, azole-resistant Aspergillus species [75,76] and azole-resistant Candida species [77] have become a problem. By standardizing the target concentration with TDM, pharmacists may be able to participate not only in efficacy and safety but also in the mechanism of resistance development.

In this article, AMR Alliance Japan also suggests that efforts should be made to promote an increase in the amount of therapeutic drug monitoring (TDM) in the Japanese government’s next NAP on AMR [8]. Since the author is one of the members of the TDM guidelines board in Japan, we have summarized support for the antimicrobial stewardship program centered on TDM.

## 4. Conclusions

We reported on recent AMR strategies and studies conducted by Japanese pharmacists in the context of the COVID-19 pandemic. Pharmacists are an integral part of the antimicrobial stewardship team and must be actively involved in antimicrobial management. In Japan, the titles of board-certified infection control pharmacy specialist and infectious disease chemotherapy pharmacist exist. However, the staffing constraints that allow pharmacists to complete such programs in some hospitals in Japan and the shortage of ID-trained pharmacists are barriers to the implementation of the program and present challenges. Japanese pharmacists are working to reduce AMR and change the Japanese system while accumulating achievements one step at a time. Japanese pharmacists strive to properly utilize TDM as a tool to improve the adequacy of antimicrobial use. Further validation is needed in the future to discuss the use of a proactive TDM, defining the criteria that could identify the patients, the disease, or the conditions that are more often associated with pharmacokinetic variability. It should be noted that this review does not include a wider approach. A wider approach to antimicrobial stewardship that includes other professionals working in the ICU, infectious disease wards, cardio-surgery units, etc., for whom the possibility of spreading knowledge about the risk of multi-resistant strains and protocols to reduce the risk is of fundamental importance [78]. Pharmacists can play a role in controlling AMR by actively incorporating TDM and PK/PD theories that utilize specialized pharmacokinetics.

## Figures and Tables

**Figure 1 antibiotics-10-01284-f001:**
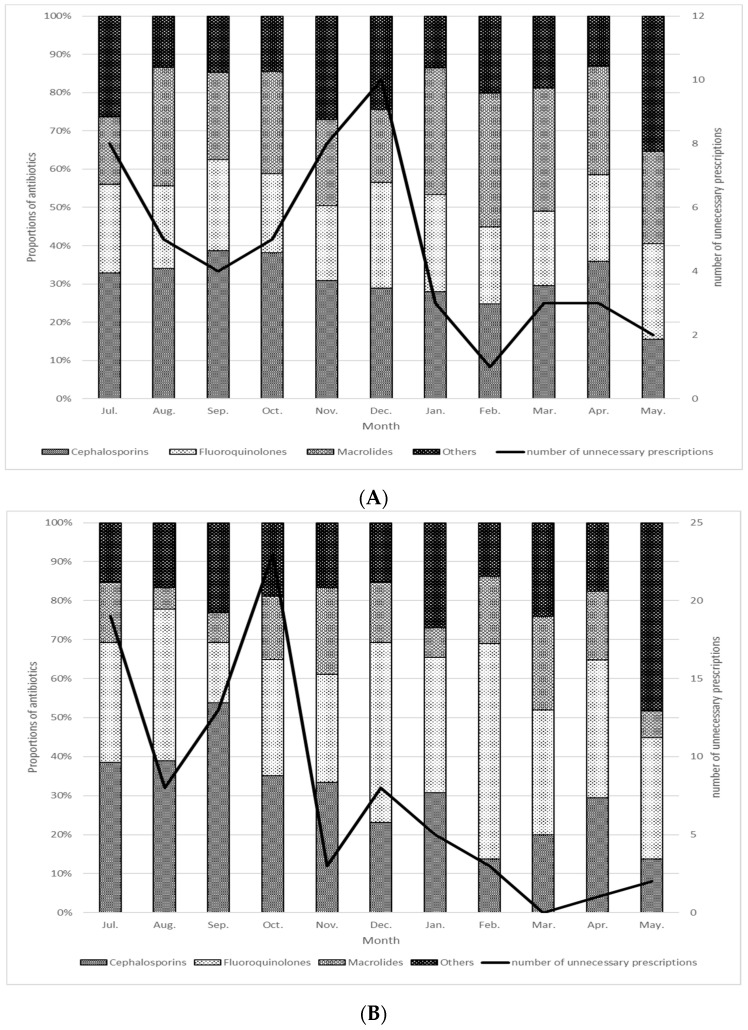
Distribution of antibiotics prescribed for acute respiratory tract infections (**A**) and acute diarrhea (**B**), 7 July 2020–May 2021 [6]. The bar graph shows cumulative percentage of oral antimicrobial agents by category. The line graph shows the change in the number of unnecessary antimicrobial agent prescriptions. (**A**) Acute respiratory tract infections, (**B**) Acute diarrhea.

**Figure 2 antibiotics-10-01284-f002:**
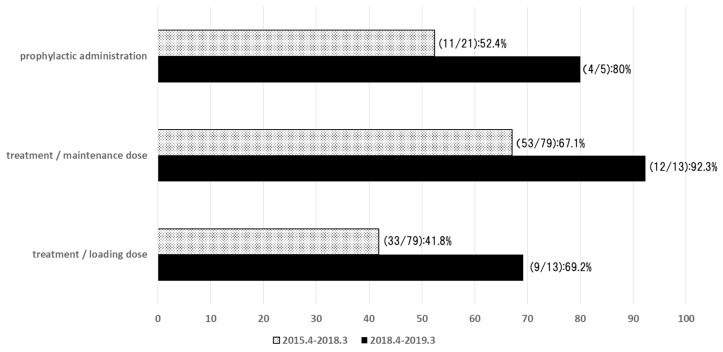
The rate of adherence to the standard dose of voriconazole. The survey was conducted at our hospital between April 2015–March 2019. Overall, data from 118 patients were reported [61].

**Table 1 antibiotics-10-01284-t001:** Suggestions for process and outcome indices for antifungal stewardship by pharmacists.

Process Index	Examples of Index
Antifungal drug consumption	Days of therapy per 1000 patient-days, defined daily doses per 1000 patient-days, or individual patients treated with antifungal drugs (excluded prophylaxis)
Compliance with institutional guidelines	Proportion of compliance using template of each facility for the following items and confirmation
Choice of drug	Proportion of patients treated with drug of choice for indication
Dose	Approved indications and dosages of each country
Administration period	For fungaemia, administration for at least 14 days after negative confirmation of blood culture
Therapeutic drug monitoring	Proportion of patients on azole and voriconazole/posaconazole for whom serum level was checked appropriately from time of initiation
Drug–drug interaction (DDI)	Proportion of patients on azole for whom DDI was checked appropriately from time of initiation
Step-down	Proportion of patients with fluconazole-sensitive Candida for whom therapy was switched from broad-spectrum agent, polyene, or echinocandin to fluconazole and intravenous to oral formulation
Use of diagnostic tests	Proportion of compliance with guideline recommendations for monitoring serum galactomannan or (1,3)-β-d-glucan or novel approaches
Source control	Proportion of patients with candidemia with catheter removal
**Outcome index**	**Examples of metric**
Treatment of invasive fungal infection	Proportion of patients with clinical cure or proportion of patients with candidemia with recurrent infection
Resistance	Proportion of Candida isolates caused by fluconazole-resistant strains
Cost	Total cost of prescriptions per year, stratified by antifungal drug

Antifungal stewardship considerations for adults and pediatrics. *Virulence*, 2017; 8(6):658–672 [58]. Partially modified by authors.

## Data Availability

All data are applicable in the paper.

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
