# Peer review of "A Strategy for Hospital Pharmacists to Control Antimicrobial Resistance (AMR) in Japan"

_antibiotics, 2021, doi:10.3390/antibiotics10111284_

Round 1
Reviewer 1 Report
The manuscript is reviewed antimicobial stewardship by pharmacists in Japan. A flow and context is excellent and the review is worth to publish the journal.
Authors used a lot of abbreviations in the manuscript. Use of abbreviation is good but there are some duplication for definition, ie MRSA. In the introduction section, there are some abbreviations those are not defined before, ie, AS and US.
I recommend for author to double check to use abbreviation in the manuscript.
Author Response
Reviewer 1
- The manuscript is reviewed antimicobial stewardship by pharmacists in Japan. A flow and context is excellent and the review is worth to publish the journal.
Respose: I’m very happy. Thank you for your wonderful comments.
- Authors used a lot of abbreviations in the manuscript. Use of abbreviation is good but there are some duplication for definition, ie MRSA. In the introduction section, there are some abbreviations those are not defined before, ie, AS and US.
Respose: Thank you for your advice. All AS have been changed to antimicrobial stewardship. And, US changed United states.
- I recommend for author to double check to use abbreviation in the manuscript.
Respose: Thank you for pointing out. Checked it systematically with two people.
Reviewer 2 Report
Introduction
- Minor English editing p1 line 34, p2 line 45 also name of figure 1, line 224
- Please provide figure 1 in greater resolution, also mention % reductions in text.
- Please put the japan practices first and then US, it is somewhat unclear written in this way
- Can you discuss the possible reasons for this difference in DALY between Japan and Europe? Are there any other modifiable factors than number of infections?
- Please rewrite the following sentence
Based on data from 14,905 inpatients at 57 hospitals, combined with data from hospital-acquired infection surveillance and claims databases, were retrospectively studied.
- This is unclear, what?
The MRSA-group-adjusted effects were 1.03-fold (95% confidence interval (CI), 1.01-1.05) and 1.04-fold (95% CI, 1.01-1.06), with an odds ratio of 1.14 (95% CI, 1.02-1.27).
- Also please write the full name when first mentioning abbreviations.
- Please provide reference for the following, possibly some guidelines
In Japan, vancomycin (VCM), teicoplanin (TEIC), arbekacin (ABK), and linezolid (LZD) had been used as four therapeutic agents against MRSA
- Also write references at the end of sentences.
- In the study by Goto please stress that included prescriptions for MRSA only if that is the case.
- Please provide a rational for the claim that this As a result, it was found that the total anti-MRSA agent use was increased significantly over time arises from the fact that there were newly approved treatments, otherwise reconsider this sentence
- In study by Kitano and another study by Niwa who developed the mentioned gudielines? were pharmacists in the teams that developed them?
- The study by Suzuki et al is irrelevant in the context of the Manuscript, please consider not mentioning it line 166, Also lines 170-184 are not relevant for this Manuscript
- Can more be said about community pharmacists and their roles? If now, please add hospital pharmacists to the title of the Manuscript
- In the mentioned studies you say were reported by pharmacists, maybe it would be better to say exactly what were the pharmacists’ roles.
- Please mention the Boards from conclusion somewhere previously in text.
- Line 402-4011 are not relevant for the subject
Author Response
Reviewer 2
- Minor English editing p1 line 34, p2 line 45 also name of figure 1, line 224
Respose: Thank you for the detailed check. Fixed it.
- Please provide figure 1 in greater resolution, also mention % reductions in text.
Respose: I tried to fix it as you pointed out. “The bar graph shows cumulative percentage of oral antimicrobial agents by category. The line graph shows the change in the number of unnecessary antimicrobial agent prescriptions.”
- Please put the japan practices first and then US, it is somewhat unclear written in this way
Respose: Thank you for your advice. US changed United states.
- Can you discuss the possible reasons for this difference in DALY between Japan and Europe? Are there any other modifiable factors than number of infections?
Respose: Thank you for your advice. Unfortunately, there was no data I could provide.
- Please rewrite the following sentence
Based on data from 14,905 inpatients at 57 hospitals, combined with data from hospital-acquired infection surveillance and claims databases, were retrospectively studied.
Respose: Thank you for your comment. I also thought this sentence was unnecessary and deleted it.
- This is unclear, what?
The MRSA-group-adjusted effects were 1.03-fold (95% confidence interval (CI), 1.01-1.05) and 1.04-fold (95% CI, 1.01-1.06), with an odds ratio of 1.14 (95% CI, 1.02-1.27).
Respose: Thank you for your advice. I added the following. L 108-110, “This result shows that disease burden was higher in inpatients with MRSA than in those with MSSA, moreover, attributable burden of methicillin resistance was significant after adjustments.”
- Also please write the full name when first mentioning abbreviations.
Respose: Thank you for pointing out. Checked it systematically with two people.
- Please provide reference for the following, possibly some guidelines
In Japan, vancomycin (VCM), teicoplanin (TEIC), arbekacin (ABK), and linezolid (LZD) had been used as four therapeutic agents against MRSA
Respose: Thank you for pointing out. Added reference (16).
- Also write references at the end of sentences.
Respose: Thank you for pointing out. Added reference (16).
- In the study by Goto please stress that included prescriptions for MRSA only if that is the case.
Respose: Thank you for your advice. Added “only” L- 117
- Please provide a rational for the claim that this As a result, it was found that the total anti-MRSA agent use was increased significantly over time arises from the fact that there were newly approved treatments, otherwise reconsider this sentence
Respose: Thank you for your very important advice. Added L-128-132
In particular, vancomycin, teicoplanin, and arbekacin were difficult to achieve the target trough concentration with the dose setting in the Japanese package insert. Adequate dosage in compliance with the guidelines may be one factor in the increase in dosage. The AMR problem may have been even more serious in Japan if the dosage were the same as before.
- In study by Kitano and another study by Niwa who developed the mentioned gudielines? were pharmacists in the teams that developed them?
Respose: Thank you for your detailed suggestions. As pointed out, it was a report from a doctor, so I deleted it.
- The study by Suzuki et al is irrelevant in the context of the Manuscript, please consider not mentioning it line 166, Also lines 170-184 are not relevant for this Manuscript
Respose: Thank you for pointing out. The full text describes the importance of TDM and I thought it was a necessary sentence. Therefore, I added a sentence about the importance of TDM by pharmacists. Added L 176-183, “Clearly, AUC monitoring of vancomycin is important. However, accurate AUC monitoring may require multiple blood samples. Therefore, it is important to properly use AUC and trough monitoring. AUC monitoring is recommended for patients at high risk of renal impairment, but for patients without AUC monitoring, it was important to build a predictable model even with conventional trough monitoring. Especially under COVID-19, we want to reduce blood sampling points as much as possible, and this result will be one tool for pharmacists to propose TDM to physicians.”
- Can more be said about community pharmacists and their roles? If now, please add hospital pharmacists to the title of the Manuscript
Respose: Thank you for your advice. Japanese community pharmacists are expected to be active in the future, but since there is almost no reference, we added hospital pharmacists to the title.
- In the mentioned studies you say were reported by pharmacists, maybe it would be better to say exactly what were the pharmacists’ roles.
Respose: Thank you for your very important advice. Added L 452-453
“ Pharmacists can play a role in control AMR by actively incorporating TDM and PK / PD theory that utilize specialized pharmacokinetics.”
- Please mention the Boards from conclusion somewhere previously in text.
Respose: Thank you for your advice. Added L. 430-434
“In this article, AMR Alliance Japan also suggests that efforts should be made to promote an increase in the number of therapeutic drug monitoring (TDM) in the Japanese government's next NAP on AMR [8]. Since the author is one of the members of the TDM guidelines board in Japan, we have summarized support for antimicrobial stewardship program centered on TDM.”
- Line 402-4011 are not relevant for the subject
Respose: Thank you for your very important advice. I thought the sentence here was important, so I added the following. Added L. 426-429, “As a recent topic, azole-resistant Aspergillus species [75, 76]. and azole-resistant Candida species [77] have become a problem. By standardizing the target concentration with TDM, pharmacists may be able to participate not only in efficacy and safety but also in the mechanism of resistance development.”